# Unlocking Resilience: How Physical Literacy Impacts Psychological Well-Being among Quarantined Researchers

**DOI:** 10.3390/healthcare11222972

**Published:** 2023-11-16

**Authors:** Han Meng, Xin Tang, Jun Qiao, Huiru Wang

**Affiliations:** 1Department of Physical Education, Shanghai Jiao Tong University, Shanghai 200240, China; maxmeng07@sjtu.edu.cn; 2School of Education, Shanghai Jiao Tong University, Shanghai 200240, China; xin.tang@sjtu.edu.cn; 3Shanghai Changning Mental Health Center, Shanghai 200335, China; qiaojun19791117@126.com

**Keywords:** physical literacy, burnout, resilience, relationship, researcher, quarantine

## Abstract

This study investigates the interplay between physical literacy, resilience, and burnout among researchers who experienced strict home quarantine during the COVID-19 pandemic in China, with a particular focus on the mediating role of resilience. Employing a two-stage sampling procedure, a total of 363 researchers from diverse disciplines, notably social science and natural science, were enlisted and administered a series of validated scales, including the Perceived Physical Literacy Scale (PPL), the 10-item Connor–Davidson Resilience Scale (CD-RISC-10), and the Chinese version of the Oldenburg Burnout Inventory (OLBL), via an anonymous online platform. The findings revealed substantial differences in physical literacy, resilience, and burnout across disciplines. Resilience partially mediated the relationship between physical literacy and burnout. Upon a closer examination of the sub-dimensions, resilience was found to fully mediate between factors of motivation, exhaustion, and disengagement. Moreover, one aspect of physical literacy—interaction with the environment—exhibited weaker correlations with both resilience and burnout compared to other dimensions of physical literacy. Overall, the study confirms the significant correlation between physical literacy and psychological parameters, establishing that elevated levels of both physical literacy and resilience serve as key factors in mitigating burnout during the pandemic.

## 1. Introduction

The subject of well-being in academic circles has garnered increasing attention in scholarly discourse over recent years. Given the highly competitive and stressful nature of academia, researchers are particularly susceptible to experiencing burnout, a chronic psychological syndrome characterized by emotional exhaustion and reduced professional efficacy [1]. Furthermore, the recent global pandemic has compounded this issue, intensifying the levels of stress, competition, and scarcity of resources experienced by researchers [2]. While the literature has extensively probed the etiology and impact of burnout in academia, less attention has been given to protective factors such as physical literacy and resilience. The concept of physical literacy, crucial for engaging in sports and physical exercises, emerges as a potential tool for enhancing well-being within the academically rigorous environment of researchers. The lack of necessary knowledge and skills could deter researchers from actively engaging in physical activities, despite their high educational background, underlining the potential benefits of enhanced physical literacy. Despite its promise, scant research has addressed the application and importance of physical literacy within an academic population. In this study, we thus aimed to understand researchers’ physical literacy levels, exploring pathways that might help them reduce their burnout.

## 2. Literature Review

### 2.1. Physical Literacy and Burnout

Understanding burnout, particularly within the demanding academic arena, is crucial for devising preventive and remedial strategies. Maslach et al. (2001) [3] defined burnout as a syndrome characterized by emotional exhaustion, depersonalization, and reduced personal achievement, primarily precipitated by long-term high emotional demands at work. Burnout further manifests as a depletion of psychological resources, inducing stress and adversely affecting mental health. Extending this understanding, Demerouti and Bakker (2008) [4] delineated burnout into exhaustion and disengagement, where exhaustion arises from physical, emotional, and cognitive stress, while disengagement reflects a withdrawal from work, work objects, and work content. Demerouti and Bakker’s exploration helps to elucidate the multifaceted nature of burnout, thereby providing a more nuanced understanding which resonates with the context of our study on the mediating role of resilience between physical literacy and burnout among quarantined researchers.

Successful recovery, facilitated by positive mental and physical recreational experiences, is underscored as a pivotal factor for maintaining high energy levels at work and mitigating burnout accumulation. Physical activity, significantly influenced by an individual’s level of physical literacy, is recognized as a substantial contributor to positive emotions, acting consequently as a buffer against burnout. Physical literacy, extending beyond the mere act of physical movement embodied by physical activity, integrates motivation, confidence, physical competence, knowledge, and understanding, fostering a lifelong commitment to physical engagement. Physical literacy encompasses three core dimensions: motivation, confidence and physical competence, and environmental interaction [5]. Motivation stimulates an individual’s enthusiasm and willingness to engage in physical activities, which in turn, boosts confidence to participate more. Physical competence, derived from active participation, not only enhances enjoyment in sports but promotes lifelong engagement, with a focus on progress rather than mastery of movement. Environmental interaction involves acquiring experiences in various sports settings, fostering social (e.g., cooperation, sharing) and cognitive development (e.g., situational awareness). Via effective engagement with environmental aspects, individuals enhance their participation in physical activities, extending these acquired abilities to other life domains, promoting an active lifestyle across different life stages. It promotes a holistic approach—mentally, physically, and environmentally—toward maintaining an active and healthful lifestyle throughout a lifetime [5].

The connection between physical literacy and physical activity is substantiated by research conducted by McKay et al. (2022) [6] and Holler et al. (2019) [7] (e.g., r = 0.51, *p* < 0.001), accentuating the positive correlation between physical literacy and physical activity. McKay et al. delve into the beneficial effects of physical literacy on overall well-being, thereby aligning with the broader notion that physical literacy, by nurturing a robust sense of self and an understanding of embodied health principles like exercise, sleep, and nutrition, markedly contributes to an individual’s emotional well-being [8]. Although this correlation has been predominantly observed in youth populations [9], it insinuates a prospective conduit for alleviating burnout among researchers, especially amid exigent epochs like the COVID-19 pandemic. These insights propel our hypothesis:

**Hypothesis 1 (H1).** 
*Physical literacy is negatively related to burnout, encompassing both exhaustion and disengagement.*


### 2.2. Physical Literacy, Resilience, and Burnout Interconnection

Building upon the concept of physical literacy, resilience is identified as a complementary attribute, albeit a multifaceted one [10]. Resilience, in a broad sense, denotes an individual’s positive adaptation and response to adversities, nurtured over time via interactions between an individual and their environment [11]. However, it is crucial to acknowledge that resilience is a multi-dimensional phenomenon with diverse interpretations across different disciplines and contexts [10]. The synergistic relationship between physical literacy and resilience becomes apparent when confronting real-life stressors; individuals fortified with both attributes are poised to navigate challenges more adeptly. As supported by Jefferies et al. (2019) [12], indicators of physical literacy are positively correlated with resilience, highlighting a synergistic relationship pivotal in reducing burnout. Hjemdal et al. (2006) [13] found that adolescents who participated in activities that required social interaction and cooperation scored higher on resilience than others. A study examining the relationship between physical activity levels and mental health in 775 Chinese adolescents showed a significant positive correlation and highlighted that physical activity that enhances resilience might be an effective way to improve adolescent physical and mental health [14]. Based on this, we propose the following:

**Hypothesis 2 (H2).** 
*Physical literacy is positively related to resilience.*


Paredes et al. [15], in their study conducted during the pandemic, elucidated that resilience significantly contributed to preserving mental health and mitigating burnout amid the crisis, with individuals possessing lower resilience facing heightened difficulties in navigating the emotional challenges brought on by the pandemic [16]. Other studies have underscored the capacity of resilience in buffering the adverse effects engendered by occupational and life stress, and its potential as a predictor of burnout [17]. In light of this, we advance:

**Hypothesis 3 (H3).** 
*Resilience is negatively related to burnout.*


Physical literacy, resilience, and burnout are correlated to a certain extent. Physical literacy, enhancing interaction with the environment and a proactive adversity response, lays the foundation for resilience, especially in challenging circumstances [11]. Resilience, pivotal in combating adversity, plays a significant role in mitigating burnout during the challenging pandemic scenario. Resilience operates according to three core dimensions: stressors, positive adaptation, and protective factors, as highlighted in the literature [16,18]. Within the physical activity ambit, factors such as disparities in individual strength, body image perceptions, and access to sports equipment are identified as potential stressors [19]. Positive adaptation reflects persistence in elevating physical activity levels to meet predefined goals, whereas protective factors—such as mastery, self-efficacy, and social skills—help mitigate adverse responses to threatening environments, cultivated by engaging in sport [20]. Continuous sports engagement fosters a proactive outlook toward new challenges, bolstering stress coping mechanisms across diverse scenario [21]. Ma et al. [22] highlighted the partial mediating role of resilience in the nexus between physical literacy and mental health. Expanded models elucidated the mediating impact of resilience on the relationship between physical literacy and broader facets of social and psychological well-being. The interrelation between burnout and aspects of physical and mental health has been substantiated [23]. Consequently, we propose that:

**Hypothesis 4 (H4).** 
*Resilience serves as a mediator between physical literacy and burnout by enhancing an individual’s ability to adapt to stressors, thereby reducing the likelihood of experiencing burnout.*


### 2.3. Variable Backgrounds: Gender and Field of Study

Research has highlighted the significant influence of individual traits and contextual factors on physical literacy, resilience, and burnout [5,24]. A notable aspect of this is gender disparities. Often, cultural norms restricting free play could lead to girls being less physically active, which, in turn, may result in reduced physical literacy and resilience [25]. Such disparities might continue into adulthood, impacting various domains of well-being.

It is posited that an individual’s physical literacy is intricately shaped by their sociocultural milieu and lived experiences [5]. Furthermore, the academic discipline one is engaged in can also significantly influence these parameters. For example, natural science researchers may find their physical activity opportunities limited due to rigid laboratory schedules, contrasting with social science researchers who, because of fieldwork and more flexible hours, might have enhanced physical literacy. The divergences in disciplinary routines extend to resilience and burnout levels. This is especially notable in regions like China, where natural sciences often receive more resources [26]. The varying academic pressures and resource allocations across disciplines could lead to different levels of physical literacy, resilience, and burnout [8,27].

It is imperative to acknowledge that these observations may not be universal across all academic cultures and institutions; they primarily serve as illustrative examples based on certain contexts. This study aims to explore these variations in depth, with a particular focus on gender and academic disciplines as key variables, to provide a more nuanced understanding of the interrelations among physical literacy, resilience, and burnout.

### 2.4. Objectives and Contributions of the Present Study

Despite the acknowledged importance of physical literacy for academics, the existing literature on this subject is limited. Past studies mainly explore the relationship between physical activity levels and either physical literacy or burnout [22]. The examination of how physical literacy correlates with psychological well-being or burnout remains underexplored. Similarly, research into the link between resilience and physical literacy is scarce [12]. Our study seeks to bridge this knowledge gap by investigating the mediating role of resilience between physical literacy and burnout, considering variables such as gender and field of study for a comprehensive understanding of these interrelationships. Specifically, we aim to investigate the following research questions:How is physical literacy associated with burnout, specifically in terms of exhaustion and disengagement, among academics?What is the relationship between physical literacy and resilience?How does resilience relate to burnout among academics, and can it act as a mediator between physical literacy and burnout?How do individual traits and contextual factors, particularly gender and field of study, influence the interrelations among physical literacy, resilience, and burnout?

## 3. Methods

### 3.1. Research Setting and Participants

The data collection was conducted in August 2022, in Shanghai, China, when residents had been requested to maintain social distance in the past two years. Following a resurgence of the epidemic, Shanghai had recently lifted a stringent four-month social restriction phase on the eve of this investigation. The sample, comprising 363 researchers (age range: 25–48, mean age: 35.5), was obtained using a two-stage sampling procedure. Initial convenience sampling was followed by snowball sampling, where initial respondents facilitated the recruitment of additional participants. The questionnaires were disseminated using a secure and anonymous online platform, and reminder emails were sent to mitigate non-response bias. A pilot test was conducted to refine the questionnaire before distribution. In this study, “researchers” refers to individuals engaged in academic research tasks within research-centric universities and research and development institutions. The gender and field of study breakdown is as provided. There were 211 males (58.1%) and 152 females (41.9%). There were 192 (52.9%) researchers in natural sciences and 171 (47.1%) researchers in social sciences. Among them, there were 42 females (24.6%) and 129 males (75%) in the field of natural sciences. There were 110 females (57%) and 82 males (43%) in the field of social sciences. All participants were fully informed of the details of the study and were free to withdraw at any time during the study, either temporarily or permanently. This study received ethical approval from the Ethics Committee of the Home Institute (Approval ID: JNUKY-2022-296I). The restricted age range was a limitation of this study, as it potentially excludes a significant portion of the academic community, particularly those aged 50 and above, who may have different levels of physical literacy, resilience, and burnout. We have referred to the official site of The Strengthening the Reporting of Observational Studies in Epidemiology (STROBE) Statement for further information: https://www.strobe-statement.org/ (accessed on 1 August 2022).

### 3.2. Measurements

Participants’ gender, education, and field of study were set as background variables, and the measurement scales were used to look at the relationship between physical literacy, resilience, and burnout.

#### 3.2.1. Physical Literacy

In order to make the scale more applicable to the research participants in mainland China, this study adapt the Perceived Physical Literacy Scale (PPL) developed by Sum [28]. Using factor analysis, PPL consisted of 13 items to measure the three dimensions of physical literacy (motivation, confidence and physical competence, and interaction with the environment). Specifically, motivation measures whether individuals would maintain positive attitudes toward physical activity throughout their life. Confidence and physical competence detected whether people could move with confidence and poise in a variety of challenging situations. Interaction with the environment assesses an individual understanding of how to move and how to relate effectively with the combined aspects of the environment in question [29]. Regarding the psychometric properties of our study, we observed good reliability (Cronbach’s alpha was 0.863, 0.858, and 0.927, respectively).

#### 3.2.2. Resilience

The resilience levels of the researchers were measured using the simplified Chinese version of the 10-item Connor–Davidson Resilience Scale (CD-RISC-10; [30]). Each item was scored on a 5-point scale from 1 (strongly disagree) to 5 (strongly agree). The higher the calculated value, the higher the toughness level. Examples of items are “I am able to adapt to change”, “I tend to bounce back after illness or hardship”, “I can handle unpleasant feelings”, etc. The CD-RISC-10 has demonstrated adequate reliability and validity of its three-factor structure in prior research reports [30].

#### 3.2.3. Burnout

To measure researchers’ burnout, the Chinese version of the Oldenburg Burnout Inventory (OLBL) was used [31]. The Chinese version of the OLBL was comprised of 2 factors (exhaustion and disengagement, including 16 items). Examples of items were “I feel more and more engaged in my work (reverse-coded)”, “Lately, I tend to think less at work and do my job automatically”, etc. The degree of agreement with each item was expressed on a 4-point ordinal scale (from 1 = totally disagree to 4 = totally agree). A higher total score indicates higher burnout. Items 1, 5, 7, 10, 13, 14, 15, and 16 were reverse-scored. The reliability of the instrument was adequate (Cronbach’s alpha was 0.933 and 0.876).

### 3.3. Data Analysis

The data analysis was meticulously carried out using IBM SPSS 26, PROCESS macro 3.5, and MPlus 8 due to their robustness in handling the required statistical methods. The analysis procedure was organized as follows: (1) Preliminary Analyses: Descriptive statistics, correlation analysis, and *t*-tests were used to identify the overall trends and characteristics, and verified assumptions of normality, linearity, and homoscedasticity; (2) Validation of Measurement Scales: Factor analysis was employed to confirm the survey’s structure validity, followed by reliability analysis (Cronbach’s α) for each factor; further validation was carried out via expert reviews and confirmatory factor analysis beyond merely reporting Cronbach’s alpha values; (3) Handling of Missing Data: Missing data were addressed using listwise deletion to ensure the robustness of the statistical analysis; (4) Mediation Analysis: Multiple linear regression was utilized to explore the relationships among physical literacy, resilience, and burnout, and examined the mediation effect using the bootstrap method with a 95% confidence interval [32]; (5) Multicollinearity Assessment: Multicollinearity was assessed using the variance inflation factor (VIF), tolerance (TOL), and Durbin–Watson statistic before proceeding with multiple linear regression; (6) Rationale for Statistical Methods and Software: The chosen methods and software were justified based on their capability to accurately analyze the relationships among the variables of interest in a cross-sectional design, addressing the generalizability concerns raised due to the sampling procedures.

### 3.4. Methodological Considerations

In this section, the methodological approach, potential biases, and the positionality of the researchers are discussed. The choice of sampling methods, the design of the questionnaire, and the potential biases inherent in the online data collection, including self-selection bias and social desirability bias, are acknowledged. The researchers’ positionality and its potential impact on the study are also reflected upon to ensure transparency and reflexivity in the research process. Furthermore, the restricted age range is recognized as a limitation that could affect the generalizability of the findings, and the absence of researchers aged 50 and above is identified as a gap that future research should address.

## 4. Results

### 4.1. Measured Variables: Descriptive Statistics and Correlation

Table 1 shows the results of descriptive statistical analysis and correlation analysis to determine the overall trend and correlation of the main variables (physical literacy, resilience, burnout, and the two dimensions of burnout).

All variables’ deviation (≤2) and kurtosis (≤4) satisfied the standard value [33], showing that the measurements were normally distributed. The result shows that resilience had significant and positive correlations with physical literacy. Burnout was significantly and negatively related to resilience and physical literacy. We also explored the correlation between the three dimensions of physical literacy and variables. Resilience was positively correlated with “motivation” and “confidence and physical competence”. However, there was a positive weak correlation between “interaction with the environment” and resilience, a negative weak correlation between burnout and “motivation”, and a significant negative correlation between burnout and “confidence and physical competence”. “Interaction with the environment” was negatively weakly correlated with burnout. The result was that the two dimensions of burnout (exhaustion and disengagement) had different degrees of negative correlation with “motivation”, “confidence and physical competence”, and “interaction with the environment”. It was worth noting that there was a very weak correlation between “interaction with the environment” and exhaustion and disengagement.

### 4.2. Differences by Gender and the Field of Study

As shown in Table 2, an independent sample *t*-test is conducted to examine the gender and field of study differences in research variables. The results show that there were significant differences in the field of study in terms of physical literacy, resilience, and burnout. Researchers in the social sciences reported higher scores in physical literacy and resilience compared to researchers in the natural sciences, with a lower score for burnout. Additionally, the gender differences were not significant to physical literacy, resilience, and burnout.

### 4.3. Multiple Linear Regression

Multiple linear regression was used to assess the association among physical literacy, burnout, and resilience. The regression model was Burnout = 4.97 − 0.21 × Physical literacy − 0.33 × Resilience. The integrated mediation model is shown in Figure 1. All correlations were strong, indicating a significant relationship among them. The explanatory power of physical literacy and resilience on burnout was 34%, which was significant. Then, we further explored the relationship between the three dimensions of physical literacy, resilience, and the two dimensions of burnout. In terms of the three core elements of physical literacy, “motivation” was positively correlated with resilience but not with burnout, exhaustion, or disengagement. “Confidence and physical competence” were positively correlated with resilience and negatively correlated with burnout, exhaustion, and disengagement. “Interaction with the environment” was not significantly correlated with any of the variables (Table 3).

In order to test the multicollinearity of multiple linear regression analysis, the variance inflation factor (VIF) and tolerance (TOL) were evaluated as test statistics. The VIF value was under 10 (1.51), and the TOL value was greater than 0.1 (0.67), both of which met the requirements. Besides, it was found that the Durbin–Watson value was 2.22, close to the standard of 2. Therefore, the possibility of autocorrelation was excluded, and the independence condition was satisfied [34].

We used the bootstrap method to assess the importance of the mediation effect (Figure 2); the significant path is represented by a solid line, and the non-significant path is represented by a dashed line. Since the 95% confidence intervals of the estimated indirect effects were not zero, the mediating effect of resilience on the relationship between physical literacy and burnout was considered to be statistically significant. Specifically, it was a partial mediator in the relationship between physical literacy and burnout. In addition, resilience was the complete intermediary between motivation and exhaustion and disengagement. Our findings suggest that resilience may significantly mediate the effect of physical literacy on exhaustion and disengagement, although other factors could also play a role. However, resilience was a partial intermediary between “confidence and physical competence” and exhaustion and disengagement. The integrated mediation model is shown in Table 4. The fit model in PLS-SEM can be analyzed from the Standardized Root Mean Square Residual (SRMR) and Normed Fit Index (NFI) values [35]. The fit degree of the structural equation model meets the relevant criteria (SRMR = 0.08, NFI = 0.69), where the recommended value of the SRMR should be less than 0.1 or 0.09, and the recommended value of NF should be between 0 and 1 [35].

## 5. Discussion

This study advances the existing body of knowledge concerning the intricate relationship between physical literacy and occupational burnout, particularly among researchers, a demographic heretofore underrepresented in the existing literature. Our findings corroborate earlier research by validating the multi-dimensional impact of physical literacy, which extends beyond mere physical well-being to encompass significant psychological variables [22]. Notably, the study augments this discourse by methodically dissecting the three fundamental components of physical literacy, examining their interplay with the dual facets of burnout, and elucidating the mediating function of resilience. Historically, academic discussions have often treated the concept of “physical literacy” monolithically, neglecting to explore its multifaceted contributions to mental health.

Furthermore, our research focuses on a high-stress occupational group, researchers, acknowledging that experiences of stress and coping strategies may vary significantly among individuals within this group due to a myriad of factors, including their specific fields of study, institutional support, and personal circumstances. Whitehead, Durden-Myers, and Pot [5] posited that physical literacy is a journey from the cradle to the grave, which is critical to happiness at every stage of life. As individuals age, they encounter a fluid landscape of environmental pressures and stressors, emanating from professional, familial, and social spheres. These challenges are especially pronounced for individuals engaged in high-stress occupations, such as researchers. Consequently, fostering physical literacy in this cohort emerges as a crucial preventative strategy for both physical and mental-health-related issues. While countries like the United States, Australia, and Canada have integrated physical literacy into their national health agendas, studies focusing on the adult population in China remain nascent. The existing literature predominantly focuses on children and adolescents, leaving a noticeable gap in understanding the implications of physical literacy for adults. Our work aims to bridge this gap, especially concerning adults engaged in professions demanding substantial cognitive effort.

### 5.1. Physical Literacy, Resilience, and Burnout Are Significantly Correlated

In line with our assumption, physical literacy is negatively related to burnout, including exhaustion and disengagement. That is, the better your physical literacy is, the less likely you are to experience job burnout. Accordingly, people will feel less exhaustion in physical, mental, and emotional aspects. Burnout is the psychological exhaustion phenomenon caused by high demands and fewer support resources at work [36]. Compared with physical tiredness and fatigue, burnout emphasizes psychological fatigue. This is also in line with previous research, which showed a positive relationship between physical literacy and positive affect and a significant relationship between physical literacy and psychological factors.

Our findings align with the second hypothesis, indicating a significant positive correlation between physical literacy and resilience, corroborating previous studies [37]. While the philosophical essence of physical literacy underscores individual–environment interactions, our data specifically highlight how physical literacy contributes to resilience. Future studies might further explore how meaningful past sports experiences influence individuals’ self-confidence and coping mechanisms across diverse environments [5]. Improving physical literacy will ensure that individuals make healthy and positive decisions throughout their lives. In addition, some studies have pointed out that the emotional dimensions (motivation and confidence) of physical literacy may help to promote mental health [38], and support adults to pursue a harmonious state between the health of the body and the mind.

The third hypothesis of the study is also supported: there is a significant negative correlation between resilience and burnout. Individuals with higher resilience can recover more swiftly from setbacks and challenges. Strict social restrictions, like those experienced during the pandemic, disrupted people’s usual study, work, and daily activities, negatively impacting their physical and mental health. For researchers facing high work pressure, factors like the blurring boundaries between work and personal life, uncertainty in policies, and stalled research progress exacerbated existing psychological stress [39]. According to previous studies, individuals with high resilience scores have shown an ability to mobilize resources to flexibly adapt to changing environments, which in turn, reduces the negative impact of social restrictions on their work and life [14]. This is also consistent with previous studies, which have shown a significant correlation between resilience and mental health, and emotional well-being dimensions [40]. From the perspective of Job demand-resources (JD-R) theory, it states that high workloads (i.e., job pressure, time pressure, work/family distractions) and lack of resources (i.e., social support, feedback, autonomy) are the most important factors leading to burnout [41]. If individuals can use various psychological resources to adapt when their work performance goals and requirements are fixed and unchangeable, they can learn to adapt to the threat of challenges and reduce the behaviors that are not conducive to adaptation.

### 5.2. Resilience Is a Mediator between Physical Literacy and Burnout

The present study empirically substantiates that resilience functions as a significant mediating variable in elucidating the association between physical literacy and occupational burnout. Initially, our findings indicate that physical literacy facilitates the development of resilience, which in turn mitigates burnout symptoms. In the realm of physical literacy, the emotional (motivation and confidence) and cognitive (knowledge and understanding) dimensions enable individuals to mobilize resources more adeptly when encountering novel challenges [42]. Concurrently, heightened levels of physical literacy correlate with robust interpersonal networks and social support, further fortifying one’s ability to withstand negative life events [43]. Furthermore, the existing literature posits that both resilience and physical literacy can be enhanced using targeted interventions [12]. As suggested by the existing literature, individuals with higher degrees of physical literacy are inherently equipped with a broad spectrum of adaptive responses, and resilience amplifies this capacity, especially in coping with adversities like epidemic conditions [14]. Theoretical considerations suggest that resilience augments the efficacy of physical literacy in counteracting burnout by mitigating the physical, emotional, and cognitive strains engendered by long-term occupational stress, as well as by attenuating negative attitudes related to job identity and job engagement [44]. The results suggest that fostering resilience using physical literacy could contribute to a more effective reduction in burnout in the work environment. Research institutions may consider organizing challenging team-building activities on a regular basis; activities such as rock climbing and wilderness exploration have been shown to help individuals develop confidence and perseverance in facing difficulties [24].

In an effort to elucidate the nuanced mediating role of resilience within the context of physical literacy and occupational burnout, this study extends the exploration to consider resilience’s mediation between three specific dimensions of physical literacy—namely, confidence, physical competence, and motivation—and the two main facets of burnout: exhaustion and disengagement. First, our analysis reveals that resilience partially mediates the relationship between both confidence and physical competence and the components of burnout. This finding aligns with the existing literature, affirming the role of confidence and physical competence as pivotal protective factors against stressors in challenging physical environments [45]. Such attributes facilitate positive individual adaptation and resilience development. Second, the data from our study hint at the possibility that resilience may act as a mediator between motivation and burnout (exhaustion, disengagement), presenting an intriguing avenue for further exploration. It suggests that solely having a positive attitude and understanding toward sports and physical activities might not be sufficient to alleviate the physical, emotional, and cognitive fatigue caused by work, or to enhance job satisfaction and recognition. Engaging actively in physical activities may foster resilience, which in turn could potentially serve as a buffer against symptoms of exhaustion and disengagement. This insight opens up a promising perspective for devising interventions to mitigate burnout among researchers, underscoring the potential value of fostering both physical literacy and resilience [45]. However, these findings warrant further validation using additional research to establish the extent of the mediating role of resilience in the interplay between motivation and burnout. Based on these findings, institutional stakeholders might consider providing opportunities and resources for researchers to translate motivational constructs into concrete actions, and recognize the potential benefits of incorporating physical activities aimed at enhancing resilience into intellectually demanding work environments.

Thirdly, our data revealed a marginal correlation between interaction with the environment and resilience and burnout. A plausible explanation for this could be that interaction with the environment might not directly impact burnout, exhaustion, and disengagement. Rather, it lays the foundational premise for the nurturing of motivation, self-confidence, and physical competence, which, in turn, might have an indirect effect on burnout. For instance, a positive experience in sports activities can foster an individual’s physical literacy, enhancing the likelihood of continued engagement in such activities. While our data did not indicate a significant direct effect of environmental interaction on burnout, it suggests a potential role in nurturing other aspects of physical literacy and promoting physical literacy as a whole [46]. Given the limited exploration of this area in previous studies, further probing into the underlying reasons behind and mechanisms of this phenomenon is encouraged.

Moreover, the mechanisms underlying resilience assessment have been thoroughly detailed in Section 2.2 of this manuscript, where resilience is explored according to its three core dimensions: stressors, positive adaptation, and protective factors. These mechanisms are pivotal in understanding the mediating role of resilience in the relationship between physical literacy and burnout.

### 5.3. Differences in Physical Literacy, Resilience, and Burnout among Researchers in Different Research Fields

Our findings tentatively suggest that there might be variations in physical literacy, resilience, and burnout among researchers from different disciplinary domains during the pandemic era. It appears that social science researchers could potentially exhibit better physical literacy and resilience while experiencing lower burnout levels compared to natural science researchers. The divergent nature of research activities inherent to each discipline could possibly underlie these observed variations. For instance, natural science researchers, engaged in fields like biology and medicine, often require specialized environments such as laboratories for conducting their research. The pandemic-induced lockdown in Shanghai necessitated a pause on lab-based investigations, despite the existing deadlines for report submissions, which might have exacerbated psychological distress among these researchers. On the other hand, the flexible nature of social science research, allowing for online data collection methods like surveys and interviews [47], might have afforded some adaptability to social science researchers, potentially mitigating the adverse effects of home-based work setups.

These observations, although preliminary, underscore the importance of further exploration into the causal factors driving the differences in levels of physical literacy, resilience, and burnout among researchers across various disciplinary fields. Particularly during pandemic-like scenarios, special attention might be warranted for natural science researchers. Additionally, the prevalent gender disparity in many natural science fields worldwide prompts further investigation [48], especially to discern the impact of such gender dynamics on the interplay between physical literacy and mental health.

## 6. Limitation

While our study provides valuable insights into the relationships between physical literacy, resilience, and burnout, it is important to acknowledge the inherent limitations. The reliance on self-report scales may introduce response bias, impacting the objectivity of the survey results. Additionally, the geographical confinement of the sample to researchers in Shanghai during a specific period may limit the generalizability of the findings to other regions or periods. The use of convenience and snowball sampling procedures may have introduced selection bias, limiting the representativeness of the sample.

Furthermore, the cross-sectional design precludes causal inference, warranting further longitudinal investigations to better understand the dynamics over time. The scope of our study could also be expanded in future research to include a broader range of occupational groups and to explore the potential moderating effects of other variables such as organizational culture or individual differences in coping strategies. Lastly, while our study hints at potential gender disparities, a more nuanced examination of gender dynamics and their impact on physical literacy, resilience, and burnout is encouraged for future research. And the lack of researchers aged 50 and above is recognized as a gap that future research should aim to fill.

## 7. Conclusions and Practical Implications

This study illuminates the interconnectedness of physical literacy, resilience, and burnout, and underscores the salutary impact of physical literacy on alleviating burnout among groups engaged in high-intensity mental work, especially amidst the epidemic prevention and control measures during the pandemic. These findings bolster advocacy for cultivating physical literacy throughout the professional lifespan of adults, thereby contributing to enhanced well-being and a mitigation of stress during high-demand situations.

The insights gleaned herein advocate for a proactive engagement in physical exercise by research personnel, or the organization of such activities by their respective institutions. Integrating these activities within the daily routine, during lunch breaks or after work hours, could serve as a catalyst in creating a conducive environment for resilience and physical growth amidst challenging occupational scenarios. Such a proactive stance not only augments individual mental resource reservoirs but sets the stage for a significant attenuation of job burnout.

The imperative for devising direct interventions, such as physical literacy programs and resilience training workshops, is accentuated by our findings. Institutions harbor the potential to spearhead these initiatives, propelling a trajectory toward enhanced physical literacy and resilience, and in turn, a reduction in burnout. This narrative transitions into policy discourse, where the evident correlations signal the integration of physical literacy into workplace policies and public health advisories. The evolution of such policy frameworks could serve as a cornerstone in mitigating burnout across high-stress professional domains.

This narrative further contemplates the economic considerations entailed in embracing physical literacy and resilience programs. A meticulous appraisal assessing the economic feasibility and potential long-term merits of these programs compared to other burnout reduction interventions could unveil cost-effective strategies for organizations and policymakers. The long-term dividends of fostering physical literacy and resilience could span improved job satisfaction, reduced absenteeism, and an escalation in productivity.

The array of practical implications outlined in this conclusion furnishes a clear yet comprehensive elucidation of the real-world applications and broader conclusions emanating from the study’s findings.

## Figures and Tables

**Figure 1 healthcare-11-02972-f001:**
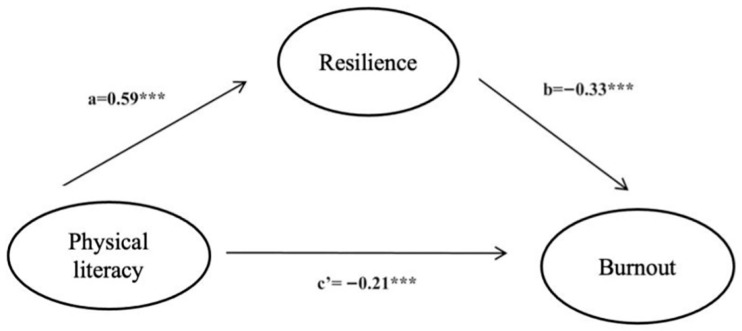
Graphical representation of the mediation model of resilience for physical literacy and burnout. *** *p* < 0.001.

**Figure 2 healthcare-11-02972-f002:**
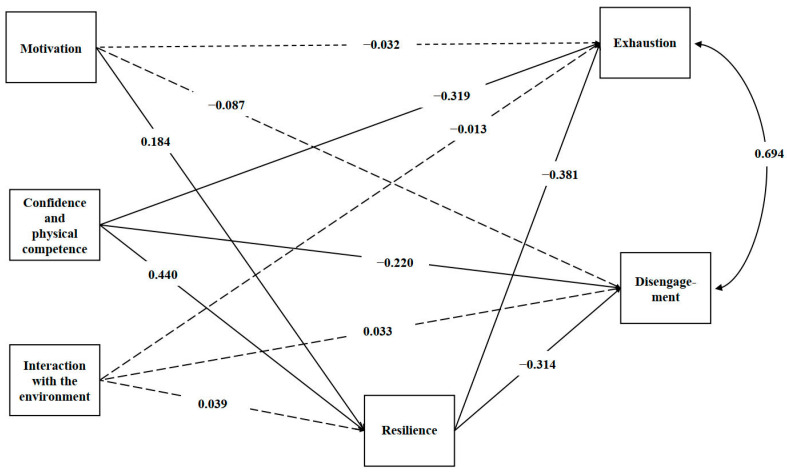
Graphical representation of the mediation model of resilience for the three dimensions of physical literacy and the two dimensions of burnout.

**Table 1 healthcare-11-02972-t001:** Descriptive statistics and Pearson’s correlations of measured variables.

	PL	PL1	PL2	PL3	Resilience	Burnout	Exhaustion	Disengagement
M	3.93	3.89	3.86	4.51	3.76	2.91	2.99	2.82
SD	0.07	0.84	0.75	0.63	0.71	0.58	0.60	0.64
Skewness	−0.45	−0.66	−0.50	−1.85	−0.27	−0.12	−0.13	−0.11
Kurtosis	−0.14	−0.04	0.26	1.25	0.31	1.01	0.85	1.04
PL	1	0.93 ***	0.87 ***	0.63 ***	0.58 ***	−0.48 ***	−0.47 ***	−0.43 ***
PL1		1	0.68 ***	0.50 ***	0.50 ***	−0.40 ***	−0.38 ***	−0.37 ***
PL2			1	0.40 ***	0.58 ***	−0.51 ***	−0.52 ***	−0.45 ***
PL3				1	0.31 ***	−0.23 ***	−0.24 ***	−0.19 ***
Resilience					1	−0.55 ***	−0.54 ***	−0.48 ***
Burnout						1	0.95 ***	0.94 ***
Exhaustion							1	0.89 ***
Disengagement								1

Note: PL = Physical literacy, PL1 = Motivation, PL2 = Confidence and physical competence, PL3 = Interaction with the environment, *** *p* < 0.001.

**Table 2 healthcare-11-02972-t002:** Comparison of gender and field of study scores in research variables.

Category	M	SD	t
**Physical literacy **			
Male	3.94	0.68	0.268
Female	3.90	0.72	
Natural science	3.66	0.68	−6.363 *
Social science	4.20	0.59	
**Resilience **			
Male	3.81	0.70	1.424
Female	3.68	0.72	
Natural science	3.63	0.77	−2.361 *
Social science	3.86	0.67	
**Burnout **			
Male	2.87	0.55	−1.165
Female	2.96	0.63	
Natural science	2.98	0.56	−1.322 *
Social science	2.42	0.60	

Note: * *p* < 0.05.

**Table 3 healthcare-11-02972-t003:** Multiple linear regression of the relationship among different variables.

Outcome Variable	Predictive Variable	*R* ^2^	Standardized Β	SE
Burnout	Resilience	0.34 ***	−0.33 ***	0.05
	Physical Literacy	0.34 ***	−0.21 ***	0.05
	Resilience		−0.31 ***	0.05
	PL1		−0.02	0.05
	PL2		−0.22 ***	0.06
	PL3		0.01	0.05
Exhaustion	Resilience	0.34 ***	−0.37 ***	0.06
	Physical Literacy	0.37 ***	−0.21 ***	0.06
	Resilience		−0.35 ***	0.05
	PL1		−0.03	0.05
	PL2		−0.32 ***	0.06
	PL3		−0.02	0.05
Disengagement	Resilience	0.26 ***	−0.28 ***	0.06
	Physical Literacy	0.28 ***	−0.20 ***	0.05
	Resilience		−0.27 ***	0.06
	PL1		−0.08	0.05
	PL2		−0.22 ***	0.06
	PL3		−0.03	0.05
Resilience	Physical Literacy	0.34 ***	0.59 ***	0.05
	PL1	0.36 ***	0.15 ***	0.06
	PL2		0.41 ***	0.05
	PL3		0.04	0.05

Note: PL = Physical literacy, PL1 = Motivation, PL2 = Confidence and physical competence, PL3 = Interaction with the environment, *** *p* < 0.001.

**Table 4 healthcare-11-02972-t004:** Mediating effect of resilience on the relationship among different variables.

	Direct Effect				
	β	SE	CILL	CIIU	R(%)
PL and burnout	−0.21	0.05	−0.28	−0.11	52.5
PL1 and exhaustion	0.03	0.08	−0.15	0.23	0
PL1 and disengagement	−0.10	0.08	−0.16	0.03	0
PL2 and exhaustion	−0.43	0.07	−0.48	−0.21	64
PL2 and disengagement	−0.29	0.07	−0.41	−0.14	59.4
PL3 and exhaustion	−0.02	0.08	−0.11	0.12	0
PL3 and disengagement	0.05	0.09	−0.06	0.10	0
	**Indirect Effect**				
	**β**	**SE**	**CILL**	**CIIU**	**R(%)**
PL and burnout	−0.19	0.04	−0.31	−0.11	47.5
PL1 and exhaustion	−0.08	0.08	−0.44	−0.19	>100
PL1 and disengagement	−0.06	0.06	−0.544	−0.20	>100
PL2 and exhaustion	−0.18	0.06	−0.39	−0.17	36
PL2 and disengagement	−0.15	0.06	−0.40	−0.18	40.6
PL3 and exhaustion	−0.02	0.04	−0.06	0.02	0
PL3 and disengagement	−0.01	0.03	−0.05	0.01	0
	**Total Effect**				
	**β**	**SE**	**CILL**		**CIIU**
PL and burnout	−0.40	0.04	−0.49		−0.31
PL1 and exhaustion	−0.08	0.08	−0.66		−0.36
PL1 and disengagement	−0.06	0.08	−0.66		−0.36
PL2 and exhaustion	−0.50	0.07	−0.74		−0.49
PL2 and disengagement	−0.44	0.07	−0.69		−0.14
PL3 and exhaustion	−0.03	−0.14	0.02		0
PL3 and disengagement	0.04	−0.07	0.09		0

Note: PL1 = Motivation, PL2 = Confidence and physical competence, PL3 = Interaction with the environment. The symbol ‘β’ = the standardized coefficient, ‘SE’ = Standard Error, reflecting the statistical precision of ‘β’. ‘CILL’ = Confidence Interval Lower Limit, and ‘CIIU’ = Confidence Interval Upper Limit, which are the lower and upper bounds, respectively, of the 95% Confidence Interval for ‘β’.

## Data Availability

The data presented in this study are available on request from the corresponding author.

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
