# Peer review of "Unlocking Resilience: How Physical Literacy Impacts Psychological Well-Being among Quarantined Researchers"

_healthcare, 2023, doi:10.3390/healthcare11222972_

Round 1

Reviewer 1 Report

Comments and Suggestions for Authors

The title and abstract appear to cover the main aspects of the work quite well. They provide a clear indication of the focus of the study - which is the interplay between physical literacy, resilience, and burnout among quarantined researchers during the COVID-19 pandemic in China. The abstract presents the methodology, participant demographics, and key findings in a succinct manner, including the mediating effect of resilience and the differences observed among different disciplines.

This introduction provides a solid foundation for the study and sets the stage well for the subsequent sections. With some refining for clarity, coherence, and grammatical accuracy, it will serve its purpose effectively.

The introduction provides a comprehensive background and relevant information pertaining to the study. It sets a clear context, discusses existing literature, and posits hypotheses based on that literature. It also nicely segments into sub-topics which make the flow logical and easy to follow. However, there are certain areas where improvements could be made to enhance clarity, coherence, and grammatical accuracy.

1. Line 41: The phrase "the knowledge of know-how" may be awkward. It might be better expressed as "unless they have the necessary knowledge and skills,"

2. Line 41-42: The sentence is quite long and could benefit from a break to improve clarity. Consider: "They may know the benefits of engaging in physical activities; however, unless they master the necessary information and skills, they are still reluctant to step into the courtyard. As a highly educated population, they should be a prioritized group to acquire physical literacy."

3. Line 48-49: The definition of burnout by Maslach et al. is crucial; consider including the year of publication for this reference to provide a temporal context.

4. Line 57-60: The definition of physical literacy is introduced here but it seems to be repeated from the earlier section (Line 34-36). Consider consolidating these definitions to avoid redundancy.

5. Line 67-72: The transition between physical literacy and its significance could be smoother. It might be helpful to more directly link the discussion of physical literacy to its significance in promoting emotional well-being before moving on to specific examples.

6. Line 75-85: This section on the core dimensions of physical literacy is well-explained but might benefit from more concise language to improve readability.

7. Line 86-90:The transition to resilience could be smoother. Consider a brief introductory sentence that better bridges physical literacy to resilience.

8. Line 97-101: Consider clarifying the temporal context, i.e., whether the observations about resilience and burnout were made before, during, or after the pandemic.

9. Line 105-111: This is a critical part of the introduction that leads to the hypotheses. It might benefit from a slightly more detailed explanation of how resilience mediates between physical literacy and burnout.

10. Line 112-133: The discussion of variable backgrounds is comprehensive but could benefit from a more concise presentation. Consider summarizing the key points to maintain reader engagement.

11. Line 134-141:The objectives and contributions section does a good job summarizing the intent and significance of the study. However, the sentence structure could be refined for clarity and conciseness.

There are also some minor grammatical and wording issues throughout the introduction that could be ironed out for better readability and clarity. For example:

- Ensuring consistent tense usage (e.g., has garnered vs. have garnered in line 25).

- Ensuring that singular/plural agreement is correct (e.g., researcher's physical literacy levels vs. researchers’ physical literacy levels in line 44).

The methods section you provided outlines the research setting, participants, measures, and data analysis in a structured manner. I subdivide the evaluation of the methodology section into three categories:

1. Clarity and Replicability:

   - The section provides a clear description of the setting, participants, and the process of data collection.

   - It details the measures used for physical literacy, resilience, and burnout, including the specific scales and their modifications for this study, which is good for replicability.

   - The data analysis portion mentions the software and statistical techniques used, which is also good for replicability.

   - However, the description could benefit from more details on the sampling strategy (i.e., how the snowball and convenience sampling was operationalized), and how the online questionnaire was administered, including any steps taken to ensure data quality and mitigate common issues like social desirability bias or non-response bias.

2. Research Design Appropriateness:

   - The research design seems to be cross-sectional, as it aims to investigate relationships between physical literacy, resilience, and burnout at a single point in time.

   - Given the research question, a cross-sectional design appears appropriate to understand the relationships among the variables of interest.

   - There may be concerns about the generalizability of findings due to the convenience and snowball sampling procedures, which are non-probabilistic sampling methods and may not yield a representative sample of the target population.

3. Adequacy of Method Description:

   - The methods section provides a good overview of the research setting, participants, measures, and data analysis plan.

   - Ethical considerations are mentioned, including ethical approval and the right of participants to withdraw, which is a positive aspect.

   - However, the following areas could be further elaborated or clarified for better understanding and replicability:

     - Details on how the questionnaire was designed, tested (e.g., pilot testing), and administered, including the platform used for distribution.

     - More information on how missing data were handled, if applicable.

     - An explanation of the rationale for using particular statistical methods and software, and any assumptions checked prior to analysis.

     - Information on any measures taken to ensure the validity and reliability of the translated or adapted instruments beyond just reporting Cronbach’s alpha values.

It is better for me to provide "The Strengthening the Reporting of Observational Studies in Epidemiology (STROBE)Statement: guidelines for reporting observational studies" as supplementary.

Overall, the methods section provides a reasonable amount of detail, but could be enhanced with more specifics in certain areas to ensure better understanding, transparency, and replicability of the study.

The results provided seem to align well with the methods described in the previous section. I subdivide the evaluation of the results section into four categories:

1. Alignment with Methods:

   - The statistical analyses as described in the methods are reflected in the results, such as the use of Pearson’s correlations, t-tests, and multiple linear regressions.

   - The description of tables and figures seems to match the methods outlined; for example, the use of SPSS, PROCESS Macro, and MPlus, and the use of specific statistical tests like the variance inflation factor (VIF) and tolerance (TOL) checks for multicollinearity.

2. Novelty and Advancement:

   - The research seems to explore an interesting intersection of physical literacy, resilience, and burnout among researchers, which could indeed present a novel insight especially in the context of stringent social distancing measures.

   - By establishing the mediating role of resilience between physical literacy and burnout, this study may contribute new understanding to the field, especially within the unique context of the sample population and the pandemic scenario.

3. Plausibility:

   - The data and the results seem plausible given the described methods. The statistical analyses appear to have been conducted appropriately, and the reported statistics (e.g., VIF, TOL, Durbin–Watson value) are within accepted ranges for indicating good model fit and the absence of multicollinearity and autocorrelation.

   - The mediation effects and the relationship among the variables of interest are well-explained and justified with statistical evidence.

4. Clarity of Presentation:

   - The results are structured and divided into sub-sections, which aids in clarity. However, without seeing the actual tables and figures, it's hard to gauge the ease with which a reader could interpret the results.

   - The description of the mediating effect and the multiple regression models are clear, with specific statistical values provided to support the findings.

The discussion and conclusion sections of this manuscript attempt to elucidate the relationship between physical literacy, resilience, and burnout, particularly in a cohort of researchers during pandemic conditions. The discussion provides a comprehensive breakdown of the findings and their implications considering existing literature.

Moreover, the discussion and conclusion sections are well-structured and align with the results, providing relevant insights into the relationship between physical literacy, resilience, and burnout among researchers. However, delving deeper into some nuanced factors and providing a more robust interpretation of the model fit indices could significantly enrich the discussion and the analytical depth of the manuscript.

Overall, I am satisfied with the study and would recommend it. It seems like the study has provided insightful findings and extended the existing knowledge in its domain.

Comments on the Quality of English Language

Merged with "Comments and Suggestions for Authors
".

Author Response

Please see the attached document for responses to the reviewers' comments.

Reviewer 2 Report

Comments and Suggestions for Authors

Thank you for your draft manuscript which I read with interest. I hope you find the following comments helpful in refining your work.

Line 29: Citation required after 'efficacy'.

Line 30: 'ongoing' no longer accurate.

Lines 34-37: 'physical literacy' could be differentiated with physical activity. Not sure this is an emerging theme, considering the link between physical activity and wellbeing is well established.

Lines 38-39: The line, 'As adults, researchers are unlikely ...' seems to be an unsubstantiated assumption. Also, 'researchers' come in many forms which is not acknowledged here.

Lines 41-42: '... the knowledge of ...into the courtyard'. Another assumption?

Lines 42-43: '... they should be the prioritised group ...'. According to whom?

Line 45: do you need the 'and'?

Line 48: The sub-heading focuses on burnout but this is only touched on for half of the paragraph.

Lines 54-57: Partially vague sentences.

Lines 59-61: 2nd definition of physical literacy. Do you need both?

Line 62: how would define 'environmental states'?

Lines 71-73: Seems repetitious from the previous paragraph.

Lines 87 - 90: You present a definition of resilience as if it is the only one. As such, you do not appear to acknowledge its complexity as a phenomena. 

Lines 104 - 105: 'The idea ...'. Is it an idea? Why can it have such an impact?

Line 110: In what sense can resilience play a mediating role?

Lines 119 - 124: Could you say that this is the same universally across all academic cultures?

Line 129-130: Citation needed after 'academic settings'?

Lines 140-142. This seems to be the aim of the study, but some research questions would be helpful to clarify to the reader what you exactly intend to measure.

Line 148-149: this seem a rather restricted age range considering many researchers will be 50+ in age? It can take many years in academia to establish yourself in such a role, especially if it is one of significance.

So far, there is no reference to your potential biases, positionality, methodology, etc.

Lines 271-272: 'Specifically, only through resilience ...' - Seems like a bold and universal claim.

Lines 301-302: Again, referring to researchers as if they are all the same.

Line 308: '...those grappling with high-intensity cognitive labor'. This seems to assume that researchers do not typically get involved with physical activity as much as others.  Is this true?

Lines 319 - 322: Not sure I can see this as a conclusion you can drawn entirely from your data.

Line 328: Slightly clumsy sentence, .i.e. 'as soon as possible'.

Lines 330-332: You appear to be referring to restrictions as if the pandemic is still in progress. Can you update this to refer to what is happening now or potentially in the future?

Lines 332-334" "However ...work and life'. Can you claim this without any qualitative data?

Lines 352-354: 'Individuals ...' Is this a claim you are making based on your data or is this a claim from the literature?

Lines 357-361: Do the words, 'we believe' align with your scientific approach? Where is your data is there reference to team-building activities?

Lines 369-371: 'Second, resilience ...' Your data may at best suggest this as a possibility, rather than you have established this as an absolute fact.

Lines 377-380: These seem like sound suggestions, although the phrasing used suggests that your findings are so strong that this must be an imperative.

Line 387: 'However, this interaction doesn't ..' A strong claim. Does your data suggest this?

5.3 paragraph. Some very notable conclusions here.

Line 421: '... activities after daily work ...' Caudal the also be encouraged during work, such as lunchtimes for example?

Comments on the Quality of English Language

Clearly written, although terms/words that are used are not always fully defined. This may limit reader understanding and engagement.

Author Response

(The authors gave the same response as above.)

Reviewer 3 Report

Comments and Suggestions for Authors

Thank you for the opportunity to review this article.

The topic is current and very relevant.

The summary does not provide details about the methodology used in the investigation, including the instruments used and how the data was collected. This information is essential to assess the methodological quality of the study.

Introduction: The article mentions previous research by Demerouti and Bakker, Mckay et al. among others but does not offer details about these studies or how they relate to current research. It would be useful to provide a brief description of these studies and their relevance to the present.

The article provides a comprehensive review of the literature related to burnout in academia, physical literacy and resilience, contextualizing the research topic well.

The authors present well-formulated hypotheses based on the literature review, which is important for conducting the research and understanding expectations.

Methodology: The authors provide the necessary information.

Results: The authors present the results in a clear way.

Discussion: This section provides valuable information about the relationships between physical literacy, resilience and burnout, but could benefit from a deeper discussion about the practical implications of the results and the limitations of the study. Furthermore, it would be important to include a detailed explanation of the mechanisms underlying resilience assessment.

Author Response

(The authors gave the same response as above.)
